# Process Optimization for Production of Persimmon Wine with Lower Methanol

**DOI:** 10.3390/foods13050748

**Published:** 2024-02-28

**Authors:** Jinwen Wei, Yajun Li, Yijuan Liu, Silin Liu, Xiaobing Yang, Xue Wang

**Affiliations:** 1College of Enology, Northwest A&F University, Xianyang 712100, China; 2College of Food Science and Engineering, Northwest A&F University, Xianyang 712100, China

**Keywords:** persimmon wine, low methanol, fermentation technology, methanol, pectinase

## Abstract

Persimmon wine has various nutritional elements and high commercial potential. However, the high content of methanol, which is derived from the fruit’s pectin, always hinders persimmon wine production. To reduce the methanol level in the wine, the effects of persimmon cultivar, starter, pectinase, and pretreatment methods were investigated via single-factor and orthogonal experiments. The persimmon cultivar ‘MaoKui’ was finally used throughout the study owing to its lowest pectin concentration (24.5 g/kg). The best treatment conditions against the persimmon pulp were pectinase (0.04 g/kg) at 30 °C for 4 h, then boiled at 115 °C for 15 min before fermentation started. The optimized fermentation conditions for wine production were pectinase (0.03 g/kg), 250 mg/kg starter (BO213 and SPARK with equal amounts), at 28 °C for 6 d. The obtained wine had 77.7 mg/L methanol and a 68.4% raw juice yield. The fruit wine had 111.4 mg/L methanol and a 90.6 sensory evaluation score. Forty-nine volatile aromas were identified. Ethyl acetate content was the highest, followed by 3-methyl-1-butanol, 2,3-butanediol, and lactate ethyl ester. The persimmon wine had a unique style with transparent color, elegant aroma, and pure taste.

## 1. Introduction

Persimmon (*Diospyros* spp.) is a health-promoting fruit that is rich in phenolic compounds, carotenoids, fiber, vitamins, and minerals. Over 350 species of persimmon are distributed in tropical and subtropical regions, and the global production of persimmon surpasses 5.75 million tons annually [1,2]. Due to its nutritional compositions, persimmon has a strong antioxidant capacity [3,4,5]. As one of the largest planting regions, Shaanxi produced over 330 thousand tons of persimmons in 2022 according to data released by the Bureau of Statistics of the province.

The marketing of huge amounts of fresh persimmon fruit is hindered by its limited shelf-life, and the insufficient demand results in about half of the fresh persimmon fruit spoiling in the orchards. To avoid potential economic loss, the fresh fruits can be processed in several accessible approaches to produce value-added products, such as jams, persimmon cake, juice, vinegar, and wine. The fermented persimmon beverages also contain significant quantities of organooxygen compounds, prenol lipids, fatty acyls, flavonoids, carboxylic acids, and derivatives [6]. However, persimmon beverage production is usually characterized by a poor juice yield. Owing to the presence of quantitative pectin, the crushed persimmon pulp exhibits a highly viscous state quite different from the liquid states of grapes and apples [2,7].

The use of pectinase can enhance the juice yield by decomposing the pectin, but it will significantly increase the methanol level in the resultant pulp and the final products like wine. Methanol possesses a clear anesthetic impact on the human neurological system, leading to disruptions in blood circulation within brain tissue, widespread damage to brain tissue, and potentially fatal central respiratory failure [8]. The methanol concentration in fruit wines can be affected by several aspects including the pectin content and esterification level in the fruits, the glycine metabolic activity of the chosen starter, the enzymatic hydrolysis of pectin by pectinase, and the fermentation process [9,10]. Although the fermentation process for persimmon wine production has been optimized before, the content of methanol in persimmon wine was not given special attention [2,5,6].

Considering the potential health risk, the production of persimmon wines with low methanol levels, high juice yields, and distinctive flavors is crucial for the sustainable development of the industry. In the present study, the effects of feedstock, starters, pectinase, and treatments on persimmon wine production, especially methanol levels, were investigated based on single-factor and orthogonal experiments. The results showed that the feedstock, starter, and pectinase had significant impacts on methanol production, juice yield, and sensory flavors. The results should thereby offer theoretical and technical assistance for producing favorable persimmon wine.

## 2. Materials and Methods

### 2.1. Starters and Activation

The starters used in the study were BO213, SPARK, and CH9 from Laffort Co. (European Union), bought from the local distributor. The freeze-dried starters were activated by adding 10 times distilled water (*w*/*v*), placing in a water bath at 37 °C, and stirring gently for 20 to 30 min until abundant small bubbles formed.

### 2.2. Raw Material Heat Treatment

The raw materials were treated by two different heat treatments: microwave treatment, which involved heating in a microwave (LG, Shanghai, China) oven for 3 min under medium fire (power output 350 watts, oven volume 20 L, sample volume 200 mL, with a final temperature for the microwave oven of 60–70 °C), and cooking treatment, which involved treating at 115 °C for 15 min under a steamer (LDZF-50KB-Ⅱ, Shanghai, China). When the temperature naturally decreased to 30 °C, starters were added to the persimmon pulp to initiate fermentation at 28 °C.

### 2.3. Persimmon Winemaking

Firstly, persimmon fruits from five cultivars were analyzed for their basic characteristics, and the one with the least pectin was selected for subsequent wine-making. Briefly, persimmon fruit was manually sorted, cleaned, and crushed. Then, the potassium metabisulfite (0.2 g/kg) was added to the pulp. Prior to heat treatment, the slurry was treated with pectinase (0.03 g/kg, *w*/*w*) at 30 °C for 4 h. If not stated especially, fermentation was initiated by an inoculating activated starter in 500 mL glass bottles containing 200 g of persimmon pulp throughout the present study. Sucrose was added to the pulp during vigorous fermentation. The weight loss of CO_2_ of the fermentation broth was measured at intervals of 24 h until no gas was produced. The fermentation was ended when the total sugar was below 2 g/L. The obtained persimmon wine was supplemented with sulfurous acid (1%, *v*/*v*) and stored at 4 °C for further analysis.

### 2.4. The Effects of Critical Factors on Persimmon Wine

A preliminary study was undertaken to investigate several elements that influence the sensory evaluation, methanol content, and alcohol content of fermented persimmon wine. The factors examined included fermentation temperature, sucrose addition, pectinase addition, and starter addition. The factors chosen were the starters inoculation (150, 200, 250, and 300 mg/kg), pectinase dosage (0.02, 0.03, 0.04, and 0.05 g/kg), sucrose addition (13.2, 14.9, 16.8, and 18.3%), and temperature (22, 25, 28, and 31 °C). Three starters, namely BO213, SPARK, and CH9, and their mixtures (BO213+SPARK, BO213+CH9, and SPARK+CH9) were used for persimmon wine fermentation, and their effects on sensory evaluation, methanol content, and alcohol content were evaluated. The combined yeasts were added at 75 + 75 g/kg. Then, the effects of four pectinases (P_Soleibo_, Soleibo Co., Ltd. (Beijing, China); P_Zhejiang_, Zhejiang Industrial Trade Co., Ltd. (Hangzhou, China); P_Laffort_, Laffort, Spanish; and P_Bioduly_, Bioduly Co., Ltd. (Nanjing, China) on persimmon wine were evaluated. Finally, the effects of pretreatment on persimmon wine also were investigated: 1. microwave treatment followed by pectinase hydrolyzation, 2. cooking treatment followed by pectinase digestion, 3. pectinase pretreatment by cooking treatment, and 4. pectinase by microwave treatment. Based on single-factor experiments, orthogonal experiments were performed to obtain the optimal fermentation conditions of persimmon wine.

At the end of fermentation, ethanol, methanol, and the sensory evaluation scores (SES) of the persimmon wine were measured to determine the appropriate fermentation process.

### 2.5. Analysis of Basic Physicochemical Properties 

The pH was measured using a pH meter (HI99163, Shanghai, China). The ethanol was determined using a biochemical sensor (SBA-90, Institute of Biology, Shandong Academy of Sciences, Ji’nan, China). Reducing sugars was determined by the Fehling titration method. Total acidity (TA) was titrated with 0.1 mol/L NaOH and expressed as the equivalent of tartaric acid. The methanol was determined using GC the GB/T 15038-2006 (China).

The Folin–Ciocalteu (FC) method was employed to determine the total polyphenols, with gallic acid serving as the standard compound [11,12]. Briefly, 20 µL of wine was mixed with distilled water to make 1 mL of the final volume, and then 0.1 mL of Folin–Ciocalteu’s reagent was added. After 5 min, 0.2 mL of sodium carbonate (35% *w/v*) was added. The final volume was adjusted to 2 mL with distilled water. Absorbance at 765 nm was measured in the absence of light for one hour using a suitable baseline reagent. The results were presented as mg/mL gallic acid in the wine.

The colorimetric method was used to determine flavonoids, and catechin was used as a standard [13,14]. Briefly, 0.25 mL of wine was mixed with distilled water to 1.5 mL. Then, 75 µL of 5% NaNO_2_ solution was added. After 6 min, 150 µL of 10% AlCl_3_ hexahydrate was added. It was left alone for another 5 min, and then 0.5 mL of 1 M NaOH was added. The mixture was adjusted to 2.5 mL with distilled water, mixed, and measured at 510 nm. Results are expressed by referring to the calibration curve of catechin. 

The total pectin in the persimmon wine was quantified via the McCready and McComb method [15]. Pulp (5 g) was transferred into a 50 mL centrifuge tube, followed by the addition of 35 mL of a 95% ethanol solution. The mixture was subjected to thorough agitation while being heated in a water bath at 85 °C for 10 min. Subsequently, 95% ethanol was added until the volume of the solution was approximately 50 mL. The precipitate was collected via centrifugation at 3000 r/min for 15 min; it was subsequently dissolved in a 100 mL volumetric vial containing 63% ethanol. Next, 5 mL of a NaOH solution with a concentration of 1 mol/L was added. The volume was then adjusted to the desired level using distilled water and well-mixed. The pectin compounds were isolated and then combined with carbazole under acidic conditions. The measurement of the colored solution’s absorbance was taken at a wavelength of 525 nm. The galacturonic acid (GA) was used as the standard for pectin determination, and the calibration range of GA was 20–100 g/L. Compounds of chromatographical purity from Sigma-Aldrich (Shanghai, China), including alcohols and aroma compounds such as methanol, ethanol, ethyl acetate, isobutyl acetate, isoamyl acetate, hexyl acetate, ethyl butyrate, ethyl octanoate, n-pentyl acetate, and 2-octanol, were used as an external standard for identification and quantitation of aroma compounds in persimmon wine. To be noted, if the vendor information for a chemical was not specified, it was of analytical grade and bought locally.

### 2.6. Analysis of Volatile Aromas 

The GC–MS analysis was performed as described by Li et al. [16]. GC-MS-QP2020 (Shimadzu Corporation, Shanghai, China) was coupled in series with an olfactory detector OPV 275 (Shimadzu Corporation, Shanghai, China) and a DB-WAX capillary column (60 m × 0.25 mm × 0.25 μm; Agilent J&W, Santa Clara, CA, USA). The temperature of the GC column was kept as follows: 40 °C for 3 min, increase to 160 °C at a rate of 4 °C/min, followed by an increase to 220 °C at a rate of 7 °C/min, and then kept for 10 min. Electron ionization mass spectrometric data were acquired within the mass range 35–350 *m*/*z* at 0.2 s intervals combined with the selected ion monitoring mode for quantitative analysis. Aroma compounds were identified by comparing retention times, retention indexes, aroma characteristics, and mass spectra with those of standards available in the NIST 17.0 mass spectral library. The concentration of aroma compounds was quantitated by interpolating the relative area of the sample versus the area of the internal standard using calibration curves previously established for pure standards.

### 2.7. Statistical Analysis 

All the experiments were conducted in triplicate. One-way ANOVA (Analysis of variance) and Tukey’s test with a confidence level of 95% (SPSS Inc., Chicago, IL, USA) were conducted with SPSS 25.0 software for Windows. The results were considered statistically significant when *p* < 0.05. Data were presented as mean ± standard deviation.

## 3. Results

### 3.1. Basic Characteristics of the Chosen Persimmon Fruits

The quality of wines is highly dependent on the materials [17]. To determine a suitable raw material, the five most planted persimmon cultivars in Shaanxi province were evaluated for their basic characteristics (Table 1). The total sugars and reducing sugars ranked consistently among the five persimmon varieties. Tishi persimmon exhibited the highest sugar content (33.2%, *w*/*w*), followed by Huojing and Jianshi. In contrast, Maokui and Huoguan persimmons contained the least sugars (15.4% and 14.2%, respectively). 

In terms of acidity among the persimmon fruits (Table 1), Huoguan had the highest acidity (0.29%), followed by Maokui (0.24%), Jianshi (0.22%), Huojing (0.2%), and Tishi (0.19%). The pH of the obtained pulp ranged from 5.1 to 6.1. For the total phenol contents, Tishi had the highest (46.2 mg/g), followed by Huoguan and Jianshi (both 41.2 mg/g), Maokui (10.1 mg/g), and Huojing (3.6 mg/g). The content of flavonoids ranked similarly to the total phenolic content of the persimmon varieties. The pectin contents ranged from 25.2 to 36.8 g/kg. Tishi had the highest pectin content at 36.8 g/kg, followed by Huojing (32.1 g/kg), Huoguan (27.8 g/kg), Jianshi (25.2 g/kg), and Maokui (24.5 g/kg). Considering sugar content, reduced acidity, and particularly, the methanol content, which is closely related to the pectin content, Maokui was used for subsequent experiments.

### 3.2. Starter Choice Influences Methanol Content in Persimmon Wine

The use of starters has a great influence on the wine fermentation process and the resultant wine quality [16,18,19]. Thus, three commercialized starters and their mixtures were directly inoculated into the Maoqui pulp to investigate their influence on methanol and ethanol production (Figure 1). The results showed that significant differences in alcohol levels were observed among the produced wines (*p* ≤ 0.05, Figure 1). The persimmon wine fermented by CH9 had the highest methanol content and was significantly higher than others (401.5 mg/L, *p* ≤ 0.01). The inoculation of CH9 alone or with others would enhance the generation of methanol (Figure 1). The results showed that the wine produced by BO213+SPARK yielded the lowest methanol (81.63 mg/L). The persimmon wine fermented by BO213+CH9 had the highest ethanol (7.4%, *v*/*v*), followed by BO213+SPARK, SPARK, BO213 6.3%, CH9 6.1% and SPARK+CH9 5.8%. In terms of colors, the wines showed the least difference. In aroma and taste evaluation, the wine from BO213+SPARK scored the highest for both (Appendix A). However, there were slight differences in those fermented by other starters or combinations. Especially, the wine by BO213+SPARK generated a relatively lower alcohol content. Considering the overall performance, the wine produced by BO213+SPARK displayed strong competition in color, aroma, and taste. Consequently, the starter combination BO213+SPARK was employed in the following studies. 

### 3.3. Pectinase Choice Impacts Methanol, Juice Yield and Ethanol Level

Persimmon wine’s production is retarded by the high content of pectin and the resulting low juice yield, which can be improved by the utilization of pectinase [3,9,20]. Further, the enzymatic treatment would also facilitate the extraction of antioxidant phenols from the persimmon pomace [21]. However, the enzymatic treatment of persimmon pulp usually causes the overproduction of methanol, which is toxic to people. Therefore, it is important to fine-tune the usage of the pectinase for controlling the methanol production. In the present study, four commercially available pectinases were employed to evaluate their effects on fermented wine production (Figure 2). The results showed that the methanol and juice yield were both significantly elevated (Figure 2). Among the various persimmon samples, the juice yield of persimmon wine with pectinase was significantly higher (*p* < 0.05) compared to that (55.8%) without pectinase. The wine treated by P_Soleibo_ had the lowest content of methanol (622.4 mg/L), and its juice yield was 65.0%. The wine treated by P_Laffort_ had the highest juice yield (71.9%), and its methanol content was 663.1 mg/L. Therefore, to achieve a relative balance between liquor yield and methanol content, P_Soleibo_ and P_Laffort_ were selected for further research.

### 3.4. Effect of Pretreatment on Persimmon Wine

Pretreatment strategies can influence the qualities of fruit wines [22]. The results of alcohol, sugar, methanol, liquor yield, total phenolic, and total flavonoid content of the Maokui persimmon wine obtained by adopting different treatments are shown in Table 2. Compared with the M(C)T-P1(3) wine (417.5~597.1 mg/L), the wine made with P1(3)-M(C)T had a lower content of methanol (77.7~280.1 mg/L), and the two different measures exhibited minor effects on the liquor yield. 

The liquor yield of persimmon wine made with microwave treatment (MT) was higher than that with cooking treatment (CT), but the methanol content of persimmon wine (CT) was much lower than that with microwave treatment (MT), indicating that CT was a more effective method for reducing methanol. Among wines made with P1(3)-MT, the methanol content of the persimmon wine with P_Laffort_ (P3-MT, 149.1 mg/L, Table 2) was lower than that with P_Soleibo_ (P1-MT, 280.1 mg/L), and the liquor yield of the persimmon wine made with P_Laffort_ (P3-MT) was relatively higher. Among wines, the methanol content of the persimmon wine (P_Laffort_-CT) had the lowest methanol content (77.7 mg/L), and the liquor yield of the persimmon wine (P3-CT) was relatively higher. Compared with P1(3)-MT, the liquor yield of wine made with P1(3)-CT decreased, while the total active substance content of wines with P1(3)-CT showed a great enhancement. The results above indicated that P_Laffort_-CT (pectinase was added and maintained for 4 h at 30 °C before steam treatment) was an effective method for balancing the wine yield and methanol content of persimmon wine.

### 3.5. Optimal Fermentation Conditions for Persimmon Wine

#### 3.5.1. The Effects of Starter, Pectinase and Sucrose Dosage, and Temperature

The amount of starter added can affect the quality of the wine [23]. When the starter addition is appropriate, the sugars can be efficiently converted to alcohol and other metabolites. When the starter addition is too low, the fermentation process would be greatly retarded, and increased production cost and risk of microbial contaminants can be expected. Conversely, an excessive amount of starter remaining in the wine would affect the aroma and clarity of the wine. Since the starter BO213 + SPARK had been identified as the best combination, the effects of starter dosage on the sensory evaluation score (SES) and the ethanol and methanol level of persimmon wine were investigated, with the temperature fixed at 28 °C and the pectinase addition at 0.03 g/kg (Figure 3A). The methanol content elevated significantly when the starter addition was 250 mg/kg (Figure 3A), which might be owing to the enhanced glycine metabolic activity of the starter. The ethanol content exhibited little difference (*p* > 0.05). As the amount of starter increased, the sensory evaluation scores of the persimmon wine initially rose and then decreased, reaching a maximum of 88.6 at the starter addition of 200 mg/kg (Figure 3A), which was adopted in the rest of the study.

Next, the effects of pectinase dosage on the sensory evaluation scores and the alcohol content were investigated with starter use of 200 mg/kg at 28 °C (Figure 3B). As the amount of pectinase increased, the ethanol content of the persimmon wine first increased and then slowly decreased. When the pectinase addition was 0.03 g/kg, the highest ethanol content was observed, 6.96%(*v*/*v*). The methanol content in the persimmon wine showed no obvious difference as the pectinase addition increased, indicating that 0.03 g/kg was enough, and excessive pectinase addition had no significant impact (*p* > 0.05) on the sensory properties of persimmon wines.

An appropriate ethanol level is inevitable to maintain the stability and flavor of fruit wine [24]. Considering the low sugar content of Maokui, the effects of additional sugar supplements were investigated by fixing the starter and pectinase usage at 200 mg/kg and 0.03 g/kg, respectively, at 28 °C (Figure 3C). The ethanol content in the wine increased as the total sugar elevated, and the highest ethanol content (9.42%, *v*/*v*) was obtained when the sucrose level was 18.3% (*w*/*v*). The methanol content remained almost unchanged when the total sugar was 14.9%. However, further sugar addition resulted in significant methanol elevation. The highest methanol content (227.8 mg/L) was observed when the total sugar was increased to 18.3%(*w*/*v*). In terms of SES, a remarkable difference was observed (*p* < 0.05) for sucrose addition.

The temperature has an important influence on the fermentation process via modulating yeast metabolism, enzyme activity, mass transfer, and the final quality of wine [25]. With a fixed amount of starter added of 200 mg/kg, with no sucrose addition, and with pectinase addition added of 0.03 g/kg, the results of the sensory evaluation scores and the alcohol and methanol content of the persimmon wine obtained by changing the fermentation temperature are shown in Figure 3D. The ethanol content in the tested samples slightly decreased with the fermentation temperature from 22 to 25 °C and gradually increased between 25 and 31 °C. The methanol content was 109.2 mg/L at 22 °C, the highest among the group. In summary, temperature within the tested range had obvious effects on the SES of persimmon wines.

#### 3.5.2. Optimization of Persimmon Wine Production via Orthogonal Experiments

To optimize the fermentation conditions, orthogonal experiments were subsequently conducted, based on the results of single-factor experiments with starter addition, sucrose addition, and temperature as significant influence factors (Table 3). Factor levels for orthogonal experiments were set as follows: factor A was starter addition set at four levels (150, 200, 250, and 300 mg/kg), factor B was sucrose addition (13.2, 14.9, 16.6 and 18.3%, *w*/*w*), and factor C was fermentation temperature (22, 25, 28 and 31 °C).

The orthogonal experiment results confirmed that starter usage, sucrose addition, and temperature could impact the methanol level in persimmon wine (Table 4). In terms of SES (Table 5), the most important factor was starter addition (factor A), followed by sucrose addition (factor B), and temperature (factor C). Moreover, by comparing the k1, k2, and k3 values of three factors, the best combination was A3B4C4, revealing that the optimal fermentation conditions for persimmon wine were starter addition of 250 mg/kg, sucrose addition of 10.2 g, and fermentation temperature of 31 °C. Under these conditions, the methanol content of persimmon wine was 111.4 mg/L and the sensory score of persimmon wine was 90.6.

### 3.6. Basic Chemical Characteristics and Volatile Aromas of the Persimmon Wine

Basic chemical characteristics are crucial to bookmark a specific fruit wine [26]. Therefore, we analyzed the quantitative compounds in the persimmon wine. The results showed that the final wine contained 4.4 g/L reducing sugars, 9.3% alcohol (*v*/*v*), 111.4 mg/L methanol, 79.9 mg/L SO_2_, 15.7 mg/L free SO_2_, 1338.4 mg/L phenolics, and 2684.4 mg/L flavonoids (Figure 4). All of them were within the national limits (NY/T 1508-2007 and GB/T 15037-2006, China).

Volatile aromas are important characteristics of a specific fruit wine [27]. Usually, esters, alcohols, and aldehydes are the main contributors to the aroma of fruit wine [28,29,30]. In the present study, 49 volatile aromas, including 18 esters, 17 alcohols, four ketones, three aldehydes, three acids, two phenols, one alkane, and one alkene, were detected in persimmon wine (Appendix A). Among these, ethyl acetate was the highest, followed by 3-methyl-1-butanol, 2,3-butanediol, and lactate ethyl ester. They showed different aroma characteristics and different threshold values. Certain alcohols and aldehydes like 1-hexanol and 1-nonanal emit a green fragrance. Alcohols and esters such as ethyl caprylate, isoamyl acetate, and 2-phenethyl alcohol exhibit floral aromas. Elegant aromas were formed via the combination of numerous flavoring compounds.

## 4. Conclusions

Persimmon fruit wines with lower methanol and favorable flavors are attractive. The present study showed that fruit cultivar, starter usage, and pectinase have significantly more influence on the methanol content and sensory quality of the persimmon wine than sucrose addition and fermentation temperature. Moreover, our study indicated that all the factors should be systematically investigated to produce a desirable persimmon wine. This study should provide useful insights into the processing of persimmon-derived beverages.

## Figures and Tables

**Figure 1 foods-13-00748-f001:**
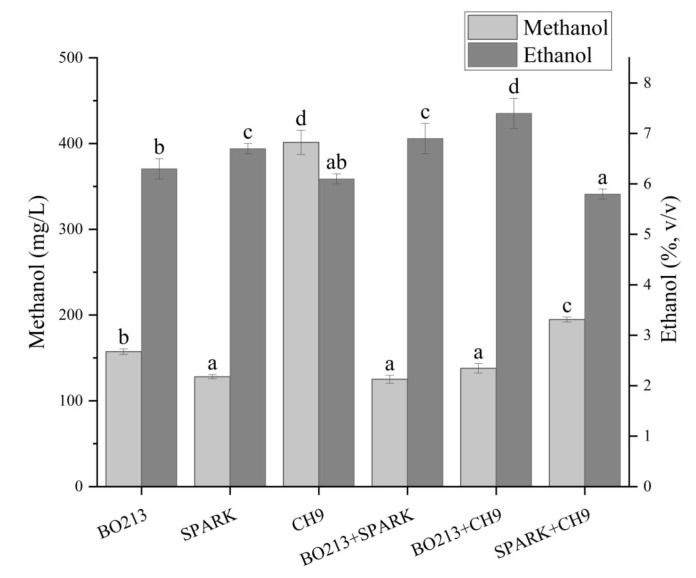
Effects of starters on methanol and ethanol in persimmon wine. Lowercase letters indicate significant differences between samples (*p* < 0.05).

**Figure 2 foods-13-00748-f002:**
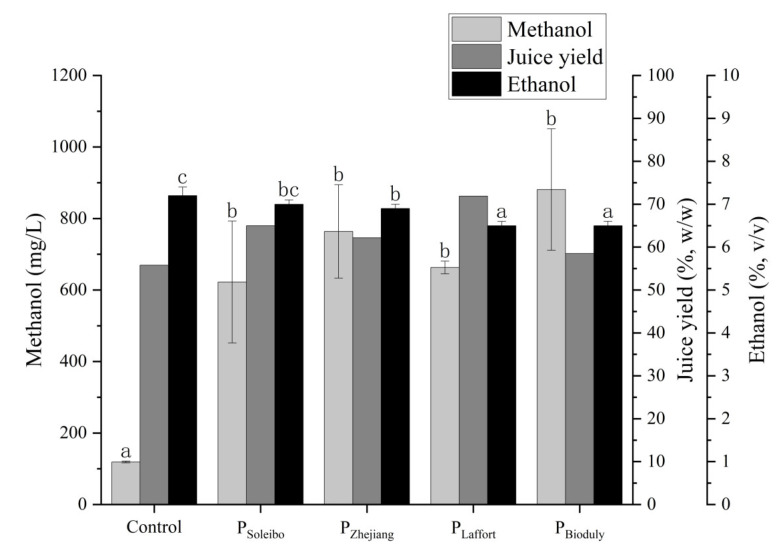
Effects of pectinase on methanol content and liquor yield in persimmon wine. Lowercase letters indicate significant differences between samples (*p* < 0.05).

**Figure 3 foods-13-00748-f003:**
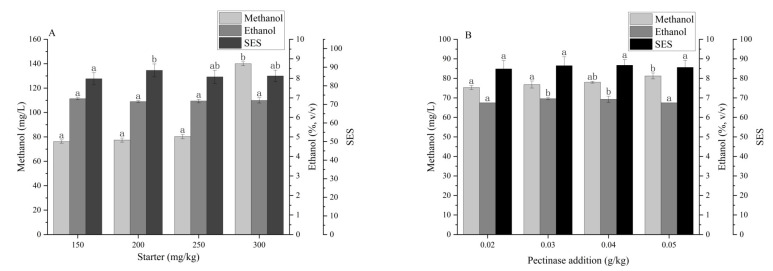
The effects of four factors on methanol and alcohol content and SES of persimmon wine. (**A**). starters; (**B**). pectinase addition; (**C**). sucrose addition; (**D**). temperature. Lowercase letters indicate significant differences between samples (*p* < 0.05). Note: It took 8 days for complete fermentation at 22 °C while 6 days were used for other groups.

**Figure 4 foods-13-00748-f004:**
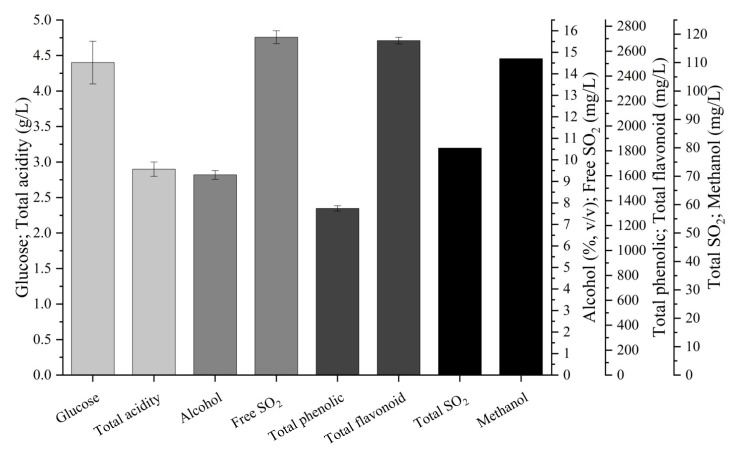
Basic characteristics of the final persimmon wine. Data were presented as means and standard derivations.

**Table 1 foods-13-00748-t001:** Basic characters of different persimmon cultivars.

Entrys	Total Acidity (%, *w*/*w*)	pH	Total Flavonoids (mg/g)	Total Phenols (mg/g)	Reducing Sugar (%, *w*/*w*)	Total Sugars (%, *w*/*w*)	Pectin (g/kg)
Huojing	0.20 ± 0.00 b	5.8 ± 0.0 d	3.0 ± 0.1 a	3.6 ± 0.0 a	22.3 ± 0.2 d	30.6 ± 1.4 d	32.1 ± 0.0 bc
Huoguan	0.29 ± 0.00 e	5.1 ± 0.0 a	49.4 ± 0.4 c	41.2 ± 0.7 d	12.8 ± 0.0 a	14.2 ± 0.6 a	27.8 ± 0.7 ab
Maokui	0.24 ± 0.00 d	5.3 ± 0.0 b	13.9 ± 0.7 b	10.1 ± 0.2 b	13.3 ± 0.3 b	15.4 ± 0.9 b	24.5 ± 0.4 a
Jianshi	0.22 ± 0.00 c	5.4 ± 0.0 c	50.0 ± 0.2 c	34.7 ± 0.5 c	19.6 ± 0.0 c	25.8 ± 1.0 c	25.2 ± 1.2 a
Tishi	0.19 ± 0.00 a	6.1 ± 0.0 e	62.9 ± 0.3 d	46.2 ± 2.0 e	22.9 ± 0.1 e	33.2 ± 2.4 e	36.8 ± 3.7 c

Notes: Values are given as the means ± standard deviations (*n* = 3), and the different letters within the same column are significantly different (*p* < 0.05).

**Table 2 foods-13-00748-t002:** Effect of pretreatment on the physical and chemical indexes of persimmon wine.

Heat Treatment	Reducing Sugar (g/L)	Ethanol (%, *v*/*v*)	Methanol (mg/L)	Juice Yield (%, *w*/*w*)	Total Phenolic (mg/L)	Total Flavonoid (mg/L)
P-Soleibo (P1)						
MT-P1	4.5 ± 0.4 c	6.8 ± 0.1 c	597.1 ± 18.4 f	65.3	268.7 ± 12.4 a	252.3 ± 4.3 a
CT-P1	6.1 ± 0.1 d	6.2 ± 0.1 b	538 ± 42.9 e	55.6	1684.7 ± 22.6 c	2456.5 ± 128.6 e
P1-MT	3.7 ± 0.2 a	6.9 ± 0.2 b	280.1 ± 29.3 c	65.0	399.6 ± 19.2 b	640.4 ± 64.3 b
P1-CT	4.7 ± 0.0 c	6.6 ± 0.1 b	135 ± 5.3 ab	58.7	2146.6 ± 46.3 e	2150.3 ± 287.3 d
P-Laffort (P3)						
MT-P3	4.2 ± 0.0 bc	6.5 ± 0.1 b	446 ± 43.7 d	68	244.8 ± 7.9 a	73.4 ± 25.7 a
CT-P3	6.3 ± 0.0 d	7.2 ± 0.1 c	417.5 ± 12.8 d	61.9	1948.8 ± 16.9 d	2929.5 ± 128.6 f
P3-MT	3.7 ± 0.4 ab	6.5 ± 0.2 a	149.1 ± 6 b	69.7	393.2 ± 25.9 b	1319.5 ± 72.9 c
P3-CT	4.5 ± 0.2 c	6.6 ± 0.1 b	77.7 ± 1.3 a	65.4	2246.3 ± 29.3 f	2926.5 ± 30.0 f

Notes: Pretreatment includes microwave treatment (MT) and cooking treatment (CT) of raw materials; P1 and P3 represent pectinase from Soleibo and Laffort, respectively. M(C)T-P1(3), pectinase 1(3) was added to fruit pulp after microwave or cooking treatment; P1(3)-M(C)T represents fruit pulp which was first treated by pectinase 1(3) and then by microwave or cooking treatment, respectively. Lowercase letters indicate significant differences between samples (*p* < 0.05). Values are given as the means ± standard deviations (*n* = 3), and the significance is expressed within the same column (*p* < 0.05).

**Table 3 foods-13-00748-t003:** Results and analysis of orthogonal experiments for persimmon wines.

No.	A	B	C	Sensory Evaluation Scores	Methanol (mg/L)
1	1	1	1	80.8	79.3
2	1	2	2	82.9	83.7
3	1	3	3	83.1	138.2
4	1	4	4	84.8	154
5	2	1	2	84.1	139.5
6	2	2	1	84	72.7
7	2	3	4	84.3	154.7
8	2	4	3	85.8	87.5
9	3	1	3	85.4	82.3
10	3	2	4	87.4	91.2
11	3	3	1	87.8	132.8
12	3	4	2	88	95.3
13	4	1	4	87.3	84.1
14	4	2	3	85.6	90.4
15	4	3	2	86	83.3
16	4	4	1	87	106.5
K1	455.2	385.2	391.3		
K2	454.4	401.8	401.8	Methanol	A4B3C1
K3	401.6	398.4	398.4		
K4	364.3	443.3	484		
R	22.725	14.525	23.175		
K1	331.6	337.6	339.6	SES	A3B4C4
K2	338.2	339.9	341		
K3	348.6	341.2	339.9		
K4	345.9	345.6	343.8		
R	4.25	2	1.05		

**Table 4 foods-13-00748-t004:** The variance analysis of orthogonal experiments with methanol content as the index.

Factor	Sum of Squares of Deviations	Free	Mean Square	F Values	*p* Values
A	1464.597	3	488.199	0.550	0.667
B	4098.467	3	1366.156	1.538	0.299
C	1428.107	3	476.036	0.536	0.675
Error	5330.634	6	888.439		

**Table 5 foods-13-00748-t005:** The variance analysis results of orthogonal experiments with SES as the index.

Factor	Sum of Squares of Deviations	Free	Mean Square	F Values	*p* Values
s	44.487	3	14.829	17.197	0.002
B	8.487	3	2.829	3.281	0.100
C	2.747	3	0.916	1.062	0.432
Error	5.174	6	0.862		

## Data Availability

The original contributions presented in the study are included in the article/Appendix A; further inquiries can be directed to the corresponding author.

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
