# Peer review of "Process Optimization for Production of Persimmon Wine with Lower Methanol"

_foods, 2024, doi:10.3390/foods13050748_

Round 1
Reviewer 1 Report
Comments and Suggestions for Authors
Line 19: I would try to keep the same naming nomenclature when it comes to chemical compounds. I.e. you use ethyl acetate the line before then switched to IUPAC lactic acid, ethyl ester. Be consistent.
What are normal yields for Persimmon wine. It's mentioned it's low but no baseline to compare it to.
How much methanol is dangerous. Again set a baseline so the reader can compare it to.
Why are you adding sugar halfway through the fermentation process and not just adjusting your starting gravity in the beginning? Is this common to persimmon wine? Brix adjustments in the wine industry are done prior to the start of fermentation.
Line 71: What characteristics did you analyses?
Line 88 - 89: In the mix fermentations did you add 150 g/kg for each yeast strain or did you add 75+75 g/kg.
Why did you not propagate the yeast prior to pitching?
Line 157 - 163: Does not need to be bold.
Table: 1: Although you do not receive any variation in the measurements you still have machine variation. The STD of pH should be the variation of the machine.
Figure 4: Since this is discussing the initial characteristics of the substrate having it in the beginning would be easier for the reader to understand the story.
VOC graphs. The graphs are hard to read because the words are so small plus the STD are hard to see. A table is more common with this type of data to see.
Comments on the Quality of English Language
I would have someone read over it. There were sections that need work in regards to the English.
Author Response
Comment 1: Line 19: I would try to keep the same naming nomenclature when it comes to chemical compounds. I.e. you use ethyl acetate the line before then switched to IUPAC lactic acid, ethyl ester. Be consistent.
Response: Thank you for your suggestion. We have revised the manuscript accordingly.
Comment 2: What are normal yields for Persimmon wine. It's mentioned it's low but no baseline to compare it to.
Response: Thank you for your suggestion. Generally, the juice yield would be around 50% (v/w) if no pectinase used.
Comment 3: How much methanol is dangerous. Again set a baseline so the reader can compare it to.
Response: Thank you for your interesting suggestion. Currently, there is no critical requirement for the content of methanol in fruit wines. As a reference, the methanol should be controlled no more than 250 mg/L for the white wine (GB 15037-2006).
Comment 4: Why are you adding sugar halfway through the fermentation process and not just adjusting your starting gravity in the beginning? Is this common to persimmon wine? Brix adjustments in the wine industry are done prior to the start of fermentation.
Response: Thank you for your question. A high initial sugar concentration would inhibit the yeast growth, and result in a longer lag phase. During the logarithmic growth phase, the yeast can metabolize sugar efficiently into alcohols and other products, so sugar is added at this stage.
Comment 5: Line 71: What characteristics did you analyses?
Response: Thank you for question. The characteristics analyzed in this study were included in Table 1. They were total acid, pH, total flavonoids, total phenols, reducing sugar, total sugar and pectin content.
Comment 6: Line 88 - 89: In the mix fermentations did you add 150 g/kg for each yeast strain or did you add 75+75 g/kg.
Response: Sorry for the vague expression. The yeasts were added at 75+75 g/kg, and the information has been provided in the revised manuscript.
Comment 7: Why did you not propagate the yeast prior to pitching?
Response: The starters are produced in a ready-to-use approach. In the present study, they were activated according to the user protocol, please refer to Part 2.1 for detailed information.
Comment 8: Line 157 - 163: Does not need to be bold.
Response: The problem has been fixed.
Comment 9: Table: 1: Although you do not receive any variation in the measurements you still have machine variation. The STD of pH should be the variation of the machine.
Response: Thank you for your comments. Table 1 has been revised.
Comment 10: Figure 4: Since this is discussing the initial characteristics of the substrate having it in the beginning would be easier for the reader to understand the story.
Response: Thank you for your suggestion, but sorry for our misleading description. The data shown in Fig. 4 was obtained from the final wine.
Comment 11: VOC graphs. The graphs are hard to read because the words are so small plus the STD are hard to see. A table is more common with this type of data to see.
Response: Thank you for your suggestion. The problem has been fixed. We have deleted Fig. 5 and contained all the related data in Supplementary table 1.
Reviewer 2 Report
Comments and Suggestions for Authors
1. at pag. 1, line 24 please correct the format as recommended by International Committee on Systematics of Prokaryotes (ICSP) and switch at (Diospyros spp.) to italicize text for genus and species;
2. at pag. 2, 2.2. Raw Material Heat Treatment, lines 65-68 – please revise completely the statement and ad more information on the final temperature for microwave oven, oven volume, sample volume and microwave generator power output during treatment;
3. at pag. 2, line 75 – please renounce at w/w for pectinase, is redundant;
4. at pag. 3, 2.6. Analysis of Volatile Aromas – please specify if columns are in coupled in series or split parallel separation in to the injector and re-join the outlet to the OPV/MS interface or analysis is done on separated column configurations;
5. at pag. 4, Table 1 - please correct the acidity of fruits or present the methodology for Total Acidity (%, w/w);
6. at pag. 7, lines 249-254 - please revise the statements;
7. at pag. 8, line 305 - please revise the statements for table 4 or SES;
8. at pag. 9, lines 326-327 - please provide method of quantification of each compound in table S1 if ware use internal standard or external for each of the identified compounds and explanation for presence in the wine samples produced.
Comments on the Quality of English Language
no comment
Author Response
Comment 1: at pag. 1, line 24 please correct the format as recommended by International Committee on Systematics of Prokaryotes (ICSP) and switch at (Diospyros spp.) to italicize text for genus and species;
Response: Thank you for your suggestion. The format has been corrected.
Comment 2: at pag. 2, 2.2. Raw Material Heat Treatment, lines 65-68– please revise completely the statement and ad more information on the final temperature for microwave oven, oven volume, sample volume and microwave generator power output during treatment;
Response: Thank you for your suggestion. The detailed information has been added (lines 67-69).
Comment 3: at pag. 2, line 75 – please renounce at w/w for pectinase, is redundant;
Response: The problem has been fixed.
Comment 4: at pag. 3, 2.6. Analysis of Volatile Aromas – please specify if columns are in coupled in series or split parallel separation in to the injector and re-join the outlet to the OPV/MS interface or analysis is done on separated column configurations;
Response: Thank you for your suggestion. The related information has been specified (line 142-145).
Comment 5: at pag. 4, Table 1 - please correct the acidity of fruits or present the methodology for Total Acidity (%, w/w);
Response: Thank you for your suggestion. Detailed description of the measurement method for total acid (TA) has been provided in 2.5 Analysis of Basic Physicochemical Properties (line 107-109).
Comment 6: at pag. 7, lines 249-254 - please revise the statements;
Response: Thank you for your suggestion. This part has been revised (line 253-259).
Comment 7: at pag. 8, line 305 - please revise the statements for table 4 or SES;
Response: Thank you for your suggestion. This part has been revised (line 307-310).
Comment 8: at pag. 9, lines 326-327 - please provide method of quantification of each compound in table S1 if ware use internal standard or external for each of the identified compounds and explanation for presence in the wine samples produced.
Response: Thank you for your useful suggestion. The information has been added to supplementary table 1.
Reviewer 3 Report
Comments and Suggestions for Authors
· First of all, please pay attention to grammatical issues. For example, don’t start a paragraph with “however” as in the second paragraph of the Introduction section.
· Have studies on this subject been conducted before? This issue has not been mentioned. Please clarify in the Introduction.
· What is the originality of the work? The originality should be strongly emphasized in the Introduction section.
· Vendor information for instruments and chemicals must be given consistently and completely: (company and country name). Some of them were missing.
· Why was the paragraph in lines 157-163 in page 4 written in bold? Any reason?
· In line 182, the authors stated “There were significant differences in alcohols levels among the produced wines. They should explain with reference to the statistical result.
· Conclusions
do not repeat what was analyzed and what were the results (conclusion means something else than summary)
clearly indicate what originates from synthesis of your results and whether the research hypotheses tends to be confirmed or not (clarify the changes for the industry/humanity/economy/environment)
Author Response
Comment 1: First of all, please pay attention to grammatical issues. For example, don’t start a paragraph with “however” as in the second paragraph of the Introduction section.
Response: Thank you for your suggestions. The grammatical issues have checked throughout the manuscript.
Comment 2: Have studies on this subject been conducted before? This issue has not been mentioned. Please clarify in the Introduction.
Response: Thank you for your constructive suggestions. Related information has been added in the “Introduction” part (line 48-50).
Comment 3: What is the originality of the work? The originality should be strongly emphasized in the Introduction section.
Response: Thank you for your constructive suggestions. Related information has been added in the “Introduction” part (line 53-55).
Comment 4: Vendor information for instruments and chemicals must be given consistently and completely: (company and country name). Some of them were missing.
Response: Thank you for your suggestion. Related information has been added (line 135-136).
Comment 5: Why was the paragraph in lines 157-163 in page 4 written in bold? Any reason?
Response: Sorry for the mistake. This part has been changed to standard format.
Comment 6: In line 182, the authors stated “There were significant differences in alcohols levels among the produced wines. They should explain with reference to the statistical result.
Response: Thank you for your suggestion.The explaination was revised by referring to the statistical result.
Comment 7: Conclusions do not repeat what was analyzed and what were the results (conclusion means something else than summary) clearly indicate what originates from synthesis of your results and whether the research hypotheses tends to be confirmed or not (clarify the changes for the industry/humanity/economy/environment)
Response: Thank you for your constructive comments. The conclusion part has been revised.